# Frequency of *Toxoplasma gondii* and Risk Factors Associated with the Infection in Stray Dogs and Cats of Panama

**DOI:** 10.3390/microorganisms8060927

**Published:** 2020-06-19

**Authors:** Lorena Fábrega, Carlos M. Restrepo, Alicia Torres, Diorene Smith, Patricia Chan, Dimas Pérez, Alberto Cumbrera, Zuleima Caballero E.

**Affiliations:** 1Centro de Biología Celular y Molecular de Enfermedades, Instituto de Investigaciones Científicas y Servicios de Alta Tecnología (INDICASAT-AIP), Panamá 0801, Panama; louiefabrega@gmail.com (L.F.); crestrepo@indicasat.org.pa (C.M.R.); 2Departamento de Clínicas y Cirugías Veterinarias, Facultad de Medicina Veterinaria, Universidad de Panamá, Campus Harmodio Árias Madrid (Curundu), Panamá 4, Apartado 3366, Panama; alicia.torresm@up.ac.pa; 3Sistema Nacional de Investigación–Secretaría Nacional de Ciencia, Tecnología e Innovasión (SNI-SENACYT), Ciudad del Saber (Clayton), Apartado 0816-02852, Panama; 4Complejo Hospitalario Veterinario de Corozal, Corregimiento de Ancón, calle Hospital, edificio 6553, 6554, 6555, Panamá 4, Apartado 3366, Panama; 5Parque Municipal Summit, Corregimiento de Ancón, Avenida Gaillard, Zona 1, Apartado 503, Panama; dsmithc@gmail.com; 6Fundación Spay Panamá. Bethania, Apartado 0818-00423, Panama; pchan@spaypanama.org; 7Centro Medico Veterinarius, Apartado 08001, Panama; centroveterinarius@hotmail.com; 8Unidad de Sistema de Información Geográfica de la Dirección de Investigación y Desarrollo Tecnológico, Instituto Conmemorativo Gorgas de Estudios de la Salud (Calidonia), Apartado 0816-02593, Panama; acumbrera@gorgas.gob.pa

**Keywords:** *Toxoplasma gondii*, Panama, infection frequency, stray dogs and cats, environment contamination, transmission, immunosuppressive diseases, oocyst

## Abstract

Stray animals such as dogs and cats have an important role in maintaining the transmission cycles and dissemination of *Toxoplasma gondii*. Therefore, the objective of this study was to evaluate the frequency of *T. gondii* in stray dogs and cats in six different regions of Panama and determine risk factors associated with the dynamics of infection in each of the studied regions. Data were obtained using serological tests for the detection of anti-*T. gondii* IgG and IgM antibodies. The results of this study revealed an overall infection frequency of 23.73%. The infection frequencies found in dog and cat populations were 25.70% and 21.93% respectively, showing no statistically significant difference. Risk factor correlations suggested different infection dynamics depending on the region analyzed. The San Miguelito, North and West regions were more associated with positive cases in dogs with an age range greater than 13 months. Conversely, the Metro, Central and East regions were more associated with negative cases in cats with age ranging between 0 and 5 months. Infection of the parasite in stray animals can be influenced by intrinsic characteristics of each region, which can potentiate different risk factors associated with the different routes of transmission.

## 1. Introduction

*Toxoplasma gondii* is one of the best adapted and most prevalent parasites in the world. It has a wide geographical distribution due to its ability to infect and multiply asexually in a great diversity of vertebrate hosts, such as birds, mammals and reptiles [1]. However, its sexual phase can only be performed in the epithelial cells of the intestines of feline species [2]. In recent decades, this parasite has aroused special interest due to the serious pathologies that *T. gondii* infection can cause in humans from the congenital and chronic point of view [3,4]. In addition, *T. gondii* is an opportunistic parasite capable of causing severe pathologies in immunocompromised populations [5].

Some authors support the hypothesis that humans became infected with *T. gondii* when the domestication of animals occurred [6]. Interaction with domestic animals since ancient times promoted the emergence and maintenance of different transmission cycles of the parasite [6]. These transmission cycles have also been closely linked to the behaviors of different populations, environmental conditions and the evolutionary capabilities of the parasite. *T. gondii* uses various routes of transmission, the most important being the ingestion of food contaminated with oocysts or tissue cysts [7]. Of all the forms of transmission used by the parasite, this seems to be the simplest and most efficient at infecting a large number of hosts. The ingestion of cyst-contaminated meat from pigs, chicken, sheep and lamb is very common in countries with high frequencies of consumption of these animals and that have preferences for undercooked meat [8].

Other forms of transmission related to the mechanical interaction with domestic animals have been highlighted mainly in dogs and cats due to their close relationship with humans [9,10]. In dogs, the risk of transmission of the parasite has been associated with their coprophagic habits and rolling behavior over grass and/or feces possibly contaminated with parasite oocysts. The oocysts can be transported in the animal’s fur and transmitted by physical contact to the human. However, studies conducted so far in dogs are inconclusive and hardly suggest a potential risk of *T. gondii* infection through mechanical contact [11,12]. In the case of felines, the greatest risk occurs when cats are infected for the first time with the parasite; without prior developed immunity to protect them, the parasite quickly infects the cells of the small intestine and begins an asexual multiplication phase. After several asexual stages, the parasite starts the sexual reproduction phase, producing millions of oocysts which will be released into the environment in the cat’s feces. Once released, the oocysts remain immature until optimal temperature and humidity conditions help them sporulate and become infectious [7,13]. After a primary *T. gondii* infection occurs, these animals develop immunity, making it very rare for them to resume oocyst release. Nevertheless, recent studies have shown that in some cases, cats can resume oocysts release six years after the primary *T. gondii* infection, suggesting that previously acquired immunity does not persist for a lifetime [14,15]. From the epidemiological point of view, there is no doubt that cats play an important role in maintaining transmission cycles; however, many factors such as the behavior of cats, environmental conditions and the maturation time and viability of oocysts, can make difficult the transmission of the parasite through feces of this animal. Therefore, this transmission route may not be as effective as the transmission through the ingestion of food contaminated with parasite oocysts. 

Epidemiological studies in Latin American countries have shown a high frequency of *T. gondii* infection in stray dogs and cats strongly associated with the region, the animal species and the type of diagnostic test used. Studies conducted in Brazil displayed variations in urban regions of the cities of São Paulo (dogs—50.5%, cats—40.0%), Rio de Janeiro (cats—18%) and Curitiba (dogs—48%), where a high percentage of positivity was observed in both dogs and cats [16,17,18]. In Argentina and Chile, the majority of the reports are studies carried out on pets. Different ranges of seroprevalences have been found in regions such as the Argentine Chaco (dogs—55.3%), Buenos Aires (cats—19.5% to 22.6%) and Northeast Argentina (dogs—13.1% to 23.0%) [19,20,21,22]. In Chile, the Valdivia and San Carlos regions yielded seroprevalence data of *T. gondii* infection in cats of 33.0% and 48.3%, respectively [23,24]. Another study in southern Chile showed a high prevalence for *T. gondii* infection in a population of domestic and feral cats (68%) [25]. Unfortunately, the data were not stratified; therefore, no inferences can be made about risk factors among these populations. Furthermore, in the cities of Bogota and Lima, the frequencies of *T. gondii* infection in abandoned dogs and domestic cats were 16.8% and 11.0–17.9% respectively [26,27,28]. 

In countries of the Caribbean, Central America and Mexico, few studies have been carried out on stray dogs and cats. However, high frequencies of infection in feral cats from an island of Puerto Rico (84.2%) have been reported [29]. In different regions of Mexico and Panama, seroprevalence data were only found in domestic animals, with ranges of 70.8–91.8% and 9.2–42.0% respectively [30,31,32,33]. Furthermore, a recent study carried out in regions of Panama City and Panama West reported high seroprevalences in domestic dogs and cats (30.73% of overall prevalence) without significant differences between the populations of both species [34].

Few studies have been developed to understand the role that stray cats and dogs play in the transmission of *T. gondii* in the Americas. Most studies have focused in domestic dogs and cats due to their close relationships with humans. Some of these studies have included samples of stray animals; however, the populations were not analyzed separately, which makes it difficult to assess the risk factors associated with the habitat of these animals and the transmission of *T. gondii*. Therefore, in this study, stray dog and cat populations from different urban regions of Panama City and West Panama were evaluated. The objectives were focused mainly on determining the frequency of *T. gondii* infection in stray dogs and cats and analyzing different risk factors associated with the *T. gondii* infection and maintenance of transmission cycles of the parasite.

## 2. Materials and Methods 

### 2.1. Ethics Statement

The methodology used in this study for the sampling of serum in stray dogs and cats was reviewed and approved by the Institutional Animal Care and Use Committee of INDICASAT AIP (IACUC-17-002, 4 May 2017), and carried out in accordance with the norms and procedures established by international regulations and those established by INDICASAT AIP.

### 2.2. Experimental Design

The experimental design used was a cross-sectional study in six populations of feral dogs and cats with different urban characteristics.

### 2.3. Geographic Area Studied 

This study was conducted in different regions of the province of Panama defined as: the Central (9°0′3.86″N 79°34′12.60″O, area: 29,000 m^2^), Metro (9°0′14.66″N 79°30′30.93″O, area: 14,957 m^2^), East (9°5′34.47″N 79°21′9.73″O, area: 18,332 m^2^), North (9°5′35.79″N 79°33′24.55″O, area: 102,386 m^2^) and San Miguelito regions (9°3′19.99″N 79°29′14.12″O, area: 23,826 m^2^). The West region located in the Province of West Panama (8°47′40.84″N 79°45′36.16″O, area: 103,508 m^2^) was also included. These coordinates were defined and generated using the Google Earth Pro program version 7.3.2.5776 (Google Address; 1600 Amphitheatre Parkway, Mountain View, CA, USA). The area of each of the regions studied was delimited by drawing a point from each one of the districts sampled, thereby producing a polygon. From this polygon, the coordinate from a midpoint known as the centroid was obtained. It is possible to find these coordinates in Google Earth by clicking on the “add a placemark” icon, placing the first point as latitude and the second as longitude. Regions were colored according to the mean ± standard deviation percentage of infection of all communities within each region using the GeoDa software version 1.12 (Center for Spatial Data Science—University of Chicago, Chicago, IL, USA). For visualization of the studied regions, a final map was generated using the Arcgis software version 10.1 (Environmental Systems Research Institute, Redlands, CA, USA).

The samples collected in this study come from 46 communities distributed as follows: 7 communities in the Central region, 6 in the Metro region, 6 in the East region, 9 in the San Miguelito region, 12 in the West region and 6 in the North region. Some communities within the San Miguelito and West regions showed variations in their sample size. However, this did not affect inter-regional analysis given that the spatial distributions of the samples collected in each of the six analyzed regions remained homogeneous, and that the sampling was based on the areas with the highest human population density, where feral dogs and cats have the highest number of interactions. The communities described in Table 1 and Table 2 only reflect the provenance of the samples, and it is through the confidence intervals shown in the tables that it was possible to observe the range of variation of the percentages of positivity found in each of the studied regions.

### 2.4. Data Collection and Survey Application

The information for each of the variables analyzed was obtained from a data collection sheet, which was filled out by the veterinarians in charge of sampling. Animal data included: age, weight, sex, species and origin of the animal. All this information was collected over a period of one and a half years between September 2017 and March 2019. The chronological ages of the animals were estimated, considering the growth of different types of teeth (incisors, canines, premolars and molars) for younger animals, and the degree of wear to determine the ages of adult cats and dogs. This methodology was performed following previously established protocols [35,36,37]. A survey was also applied to the people responsible for the animals at that moment, or to the people who captured these animals. The questions focused on each animal’s behavior at the time of capture (interaction with other stray animals and whether ingestion of any type of food was observed before capture). 

### 2.5. Sample Collection 

A total of 3 mL of whole blood was collected from 319 feral dogs and 351 feral cats, using the venipuncture procedure through the femoral, jugular and cephalic veins. The samples were placed in tubes of the brand BD vacutainer^®^, without anticoagulant and with a separator gel formed of silica particles that helped separate the serum from the rest of the blood components. The transport of the sample to the Diagnosis Laboratories of INDICASAT AIP was carried out in a container (mini cooler) with small ice packs. The sera samples were separated by centrifugation for further serological tests for the detection of anti-*T. gondii*, anti-feline leukemia and anti-immunodeficiency virus IgG and IgM antibodies.

### 2.6. Serological Diagnosis

#### 2.6.1. ELISA for the Detection of Anti-*T. gondii* IgG Antibodies 

The detection of IgG antibodies against *T. gondii*, in samples of stray dogs and cats, was performed through indirect enzyme-linked immunosorbent assay (ELISA), using the commercial kit ID Screen Toxoplasmosis Indirect Multi species (ID.vet Innovative Diagnostics, Grabels, France), following the manufacturer’s instructions. Microplates were read at 450 nm using a Multiskan™ FC Microplate Photometer version 1.01.14 model 2 (Thermo Fisher Scientific, Waltham, MA, USA). Sera with an optical density greater than 0.350 were considered positive as described in the technical information of the kit’s manual. 

#### 2.6.2. Immunochromatography Assay for the Detection of Anti-*T. gondii* IgM Antibodies 

The commercial test Nova^®^test (Atlas link technology Co., LTD, Beijing, China) based on sandwich lateral flow immunochromatography assay was used for the qualitative detection of anti-*T. gondii* IgM antibodies. Serum samples collected from all felines in the study were analyzed. The procedures and test results were performed and analyzed according to the manufacturer’s instructions.

#### 2.6.3. Immunochromatography Assay for the Detection Anti-Feline Leukemia (FeLV) and Immunodeficiency Virus (FIV) Antibodies 

Immunosuppressive diseases such as leukemia and feline AIDS were identified in cat sera using the Anigen Rapid FIV Ab/FeLV Ag Test kit (Bionote, Inc, Hwaseong, Gyeonggi, Korea) based on immunochromatography. This immunoassay was capable of detecting specific antibodies against the feline immunodeficiency virus and antigens secreted by the leukemia virus with high accuracy. The methodology used in this test was performed according to the manufacturer’s technical information.

### 2.7. Statistical Analysis

Prevalence values and basic descriptive statistics were calculated using built-in statistical functions of R version 4.0.0 (R Core Team, Vienna, Austria) [38]. The dimensionality of the dataset was reduced using a multiple correspondence analysis (MCA), which is an extension of correspondence analysis (CA) that allows for the evaluation of the pattern of relationships of several categorical variables using geometrical methods by locating each variable/unit of analysis as a point in a low-dimensional space. The MCA can be considered a generalization of principal component analysis (PCA) when the variables to be analyzed are categorical instead of quantitative [39]. Briefly, the data were reduced using the Burt’s method, as implemented in the R package FactoMineR version 1.34 [40]. Results were visualized using the factoextra R package version 1.0.5, which is based on ggplot2 for elegant visualization [41,42]. The relationships between variable categories are interpreted as follows: (1) variable categories that are grouped together share similar profiles and (2) negatively correlated variable categories are positioned on opposed quadrants. 

For estimation of the associations between toxoplasmosis and immunosuppressive diseases, a chi-square (χ2) test was performed using GraphPad Prism version 6.1 (GraphPad Software Inc., San Diego, CA, USA). The chi-square test was also used to determine the likelihood that the difference in conversion rates between a given variation and the baseline is not due to random chance between the regions studied. The alpha value for statistical significance was set at 0.05.

## 3. Results

### 3.1. Frequency of T. gondii in Stray Dogs and Cats of Panama City and West Panama

The seroprevalence for *T. gondii* infection in the population of dogs and cats in the total studied area (Panama City and West Panama) was 23.73% (95% CI: 20.59–27.17). No significant difference in percentage of positivity was found between dogs (25.70%, 95% CI: 21.07–30.93) and cats (21.93%, 95% CI: 17.79–26.71) when the Chi-square test was applied with an alpha cut-off value of 0.05 (X^2^ = 0.0198, df = 1, *p* = 0.888). The mean ages estimated for the dog and cat populations were 15.69 ± 1.26 and 7.33 ± 0.44 months respectively. The highest seroprevalence of *T. gondii* infection was observed in the North region (28.86%, 95% CI: 20.34–39.09) followed by the West (26.66%, 95% CI: 17.42–38.34) and San Miguelito (26.45%, 95% CI: 19.85–34.24) regions. The Central, Metro and East regions showed lower seroprevalences with values of 17.39%, 21.73% and 23.33% respectively (95% CI: 11.67–24.97; 14.81–30.60; 15.33–33.65) (Table 1) (Figure 1). Statistical analysis showed a significant difference only between the Central (17.39%) and North (28.86%) regions (X^2^ = 4.353, df = 1, *p* = 0.037). When the percentages of infection in dogs and cats were compared within each region, no significant differences were found either (Table 1).

### 3.2. Early Infection of T. gondii in Stray Dogs and Cats

The overall percentages of early infection of toxoplasmosis for dogs (2.82%, 95% CI: 1.38–5.47) and cats (7.69%; 95% CI: 5.22–11.12) for the total area studied, showed a statistically significant difference when both populations were compared (X^2^ = 7.798, df = 1, *p* = 0.0052). However, the detection of IgM antibodies in the dog population was low or null in some of the analyzed regions. In contrast, almost all regions showed recent infection in the cat population, except the West region, in which early *T. gondii* infection was only detected in dogs. The San Miguelito and North regions were the only ones that showed cases of recent *T. gondii* infection in both species, cats being the species with the highest number of early-infected individuals. A statistically significant difference between early-infected dogs (2.10%, 95% CI: 0.36–8.12) and cats (18.33%, 95% CI: 9.93–30.85) was found for the San Miguelito region (X^2^ = 12.604, df = 1, *P* = 0.0003), but not for the North region (X^2^ = 0.074, df = 1, *p* = 0.785). In the latter region, the *T. gondii* early infection rates remained similar for both species (8.62% for dogs and 10.25% for cats) (95% CI: 3.22–19.72; 3.34–25.16). A high percentage of primary *T. gondii* infection was also observed in felines of the East region with 16.66% positivity (95% CI: 8.36–29.79). In the Central and Metro regions the percentages of primary infection were lower with values of 1.06% (95% CI: 0.05–6.62) and 2.66% (95% CI: 0.46–10.17) respectively.

### 3.3. T. gondii Infection and Its Association with Immunosuppressive Diseases

Cat samples were analyzed for the presence of feline immunodeficiency virus (FIV) and/or feline leukemia virus (FeLV): 11.39% (95% CI: 8.36–15.30) of the total number of evaluated cats were positive for immunosuppressive diseases (9.68% for FIV, 0.85% for FeLV and 0.85% for FIV/FeLV) (95% CI: 6.89–13.39; 0.20–2.69; 0.20–2.69), and 5.12% (95% CI: 3.16–8.13) presented coinfection of immunosupressive viruses with *T. gondii*. Statistical analyses showed that there is a strong association between infection by *T. gondii* and these diseases (χ^2^ = 14.02, df = 4, *p* = 0.007). The Central (7.44%, 95% CI: 3.30–15.24) and Metro (9.33%, 95% CI: 4.15–18.85) regions showed the highest percentages of positive cats for both infections (mean ages of 5.58 ± 0.49 and 7.16 ± 0.92 months for each region respectively). For the East, San Miguelito and West regions, lower percentages of positivity were observed with values of 1.85%, 3.33% and 3.44% respectively (95% CI: 0.09–11.18; 0.58–12.55; 0.18–19.63). The population of cats for the North region did not show any positive case of concomitant infection. When statistical analyses were performed individually for each of the immunosuppressive diseases (FIV and FeLV), and no association was observed with *T. gondii* infection (Table 2).

### 3.4. Risk Factors Associated with T. gondii Infection in the Different Regions

A set of variables, including intrinsic characteristics of dogs and cats (age and behavior) and factors related to their environment, were evaluated through multiple correspondence analysis (MCA). This analysis showed how some regions formed clusters associated with specific variables that could be associated with transmission dynamics in the different regions studied. For example: The West, North and San Miguelito regions were more associated with positive cases in dogs (IgG_P) and with the highest age and weight ranges observed in the sampled population. Conversely, the Central, Metro and East regions showed a higher level of association with the negative cases in juvenile cats and with the lowest weight range found in the population analyzed. Furthermore, cats in the latter regions were also associated with the interaction with other stray dogs and cats (Figure 2).

## 4. Discussion

Central American countries have a humid tropical climate which favors the maintenance of soils and water sources contaminated with *T. gondii* oocysts [43]. These factors together with the interaction between domestic and wild animals ensure the maintenance of transmission cycles. Therefore, studies in stray animals inhabiting these regions could aid in determining the level of parasite oocyst contamination in urban and rural regions. Unfortunately, in these regions of Latin America, little is known about the frequency of *T. gondii* infection in stray dogs and cats. Studies carried out over the 80s and 90s in urban and rural regions of Panama City showed high prevalence in domestic animals and in humans of different ages [33,44]. Currently, there are no updated reports on the level of environmental contamination by oocysts and the frequency of infection in humans or stray animals. Therefore, in this study, we focused on determining the frequency of *T. gondii* infection in stray dogs and cats in different urban regions of Panama City and West Panama, and analyzing different risk factors associated with the *T. gondii* infection and maintenance of transmission cycles of the parasite.

The frequencies of *T. gondii* infection in the total populations analyzed of dogs and cats were 25.70% and 21.93% respectively, which may be an indicator of high oocyst environmental contamination in the entire studied area. Additionally, we can suggest that both species may be exposed to similar risk factors, since there was no statistically significant difference between their *T. gondii* infection frequencies. Possibly, these factors are linked to the ingestion of food thrown in garbage cans and rubbish on the streets. Analysis of the stratified data by region indicated some differences in the frequency of *T. gondii* infection. In this sense, almost all regions under study (Metro (21.73%), East (23.33%), San Miguelito (26.45%) and West (26.66%) had a homogeneous infection frequency in the total population of animals analyzed with no statistically significant difference. However, two regions (Central (17.39%) and North (28.86%), characterized by having substantially different socioeconomic factors, showed a statistically significant difference [45]. The percentage of positives for the entire population of stray dogs and cats was 23.73%. This value was lower than the one found in a study carried out on pets (dogs and cats) from Panama City and West Panama, which showed an overall percentage of positives of 30.73% [34]. However, when comparing the percentages by region of both studies, very similar frequencies were observed in two of the studied regions (San Miguelito and West Panama). The Central region presented some discrepancies regarding the analyzed communities; therefore, no inference could be made between the animal populations. The East region presented a clear difference between *T. gondii* infection frequencies of the two populations, showing 23.33% for stray animals and 39.56% for pets. These variations are possibly due to differences in the timing of the studies and varying environmental conditions. However, a higher percentage of *T. gondii* infection in pets may be probably related to the habits of some owners of feeding their pets with raw meat. Conversely, a study in Bangkok described a higher *T. gondii* prevalence in stray dogs and cats than those in households [46]. The frequencies of *T. gondii* infection in stray animals are generally higher than those reported in pets [7]. Nonetheless, a high prevalence in pets has been shown in studies carried out in Asia (34.61–57.14%), Latin America (26.90–95.80%) and Europe (32.80–48.40%) [32,47,48,49,50].

The multiple correspondence analysis (MCA) was a tool used in this study to analyze the pattern of relationships of several risk factors involved in the different transmission pathways that may be occurring in each of the regions under study. The results obtained through the MCA showed the association of a set of specific variable categories for some regions. In San Miguelito, North and West regions, dogs 13 months old or older (A3) and weighing less than 12 pounds were more associated with the positive cases (IgG_P) for *T. gondii* infection (Figure 2). This association suggests that the transmission of the parasite in 50% of the dogs that were positive in these regions required an exposure time equal to or greater than 13 months. The remaining percentage of positivity in dogs was distributed between the age ranges of 1–5 months (A1; 18.1%) and 6–12 months (A2; 30.9%), displaying an increase of *T. gondii* infection closely related to the age of the animal. Adult dogs have a longer exposure time to different risk factors that may increase the chances of infection and contribute to the spread of the parasite in the environment [51]. These animals can ingest oocysts which are subsequently eliminated through the feces without any process of multiplication of the parasite occurring [52]. The ingestion of oocysts is closely related to the dog coprophagic habits and to an environment highly contaminated by oocysts. In fact, several authors suggest that the frequency of *T. gondii* infection in stray dogs can help to estimate the level of environmental contamination produced by the release of oocysts in the feces of cats. This alternative strategy has been adopted because so far there are no efficient methods to measure the level of environmental contamination by oocyst [16,53,54]. Furthermore, *T. gondii* infection in dogs could also be associated with the ingestion of leftover meat thrown into the street or in trash enclosures easily accessible to these animals. Unfortunately, these regions do not have an organized garbage collection system that operates efficiently in all their communities, which increases accessibility to leftover meat, thereby potentially enhancing transmission of the parasite mainly in dogs [55]. In addition, two of these regions (San Miguelito and West Panama) have a higher density of houses by area, which could promote a greater interaction between stray animals and humans [45,56]. Therefore, dogs can keep transmission cycles active and increase the frequency of *T. gondii* infection among stray animals.

The Metro, Central and East regions showed a different *T. gondii* infection dynamic. These regions were more associated with the negative cases for *T. gondii* infection in the cat population. From the total of negative cats, 56.75% were associated with the lower age (A1 = 1–5 months) and weight (W1 = 0–5 pounds) ranges, and with the interaction with other stray dogs and cats (SAS_DC) (Figure 2). These results suggest that cats under 5 months of age seem to have a lower risk of *T. gondii* infection when compared to cats of higher age ranges (A2 = 6–12 months). Therefore, exposure time is also an important risk factor in the cat population [57]. The chances of *T. gondii* infection in cats older than 6 months possibly increase due to the hunting skills acquired and perfected over time [58,59]. An experienced cat will be able to hunt a greater number of animals of different species. Conversely, the interaction of juvenile cats with other stray dogs and cats (SAS_DC), does not seem to be an important risk factor for the transmission of the parasite in these regions, which may be indicative of a lower environmental contamination by oocysts.

Prevalence of IgM antibodies in the total populations of dogs and cats studied was 2.82% and 7.69% respectively. These antibodies can help to estimate in some individuals the time range in which the infection occurred and the phase of the disease (acute or chronic). Unfortunately, some cats do not develop a detectable IgM response, but in those with detectable IgM titers, this response can persist for up to sixteen weeks after *T. gondii* infection [13]. In this sense, the animals that were positive for IgM and negative for IgG antibodies, were possibly in the most acute phase of a primary infection with *T. gondii*. In this study, cats were the only species in which IgM**^+^**/IgG**^-^** individuals were found, representing 3.98% of the total population of cats. It is possible that those cats are releasing cysts; however, the release of oocysts can only be confirmed by fecal flotation or by molecular techniques such as PCR (polymerase chain reaction). On the contrary, the highest percentage of IgM-positive animals was found in the age range of 0–5 months (A1) for both species. Statistical analyses did not show significant differences for the percentages found between dogs (2.5%) and cats (5.98%) (data not shown). Therefore, it is possible that both species are exposed to the same risk factors at an early age.

Another risk factor analyzed in this study was the association between *T. gondii* infection and immunosuppressive diseases (FIV and FeLV) in cats. The results of the chi-square statistic (χ^2^) demonstrated a strong association between toxoplasmosis and FIV. Except for a single individual, none of the coinfected cats had detectable IgM titers against *T. gondii*. In addition, almost all coinfected cats were in the age range between 0 and 12 months. The Central and Metro regions had the highest coinfection frequencies (7.44%–10.76%) when compared to the other regions (1.85%–3.44%). It is possible that in these regions, cat populations have greater interactions between each other, which can generate a greater number of fights that promote the transmission of FIV. Studies in cats with coinfections between *T. gondii* and FIV showed a decrease in CD4^+^ and CD8^+^ lymphocytes, a decrease in B cells and reactivation of the parasite. Pneumonia and hepatic necrosis are some of the most severe pathologies reported in these animals [60]. However, the release of oocysts in these animals has not yet been well defined. In the total population of cats analyzed for *T. gondii* and immunosuppressive diseases (FIV and FeLV), the infectious agent showing the highest frequency was *T. gondii* in 21.93% of the cases, followed by FIV (9.68%) and at a lower percentage was FeLV (0.85%). These variations in the frequency of these infections may indicate the effectiveness of *T. gondii* transmission cycles and the adaptive success of this parasite in urban regions.

## 5. Conclusions

The infection frequency with *T. gondii* is similar in populations of dogs and cats that are exposed to risk factors inherent to urban environments. Therefore, differences in the natural behavior of each species do not seem to have a significant effect in increasing *T. gondii* infection. The ingestion of food waste in regions with higher availability may be an important risk factor for both species. In this sense, socioeconomic and environmental factors specific to each geographic region seem to be associated with the increase or decrease in *T. gondii* infection frequencies. Uncontrolled proliferation of these animals and the lack of a good garbage collection system are factors that can promote the spread of *T. gondii* infection in both urban and rural regions. Moreover, in both species the risk of contracting the *T. gondii* infection increases with the age of these animals.

## Figures and Tables

**Figure 1 microorganisms-08-00927-f001:**
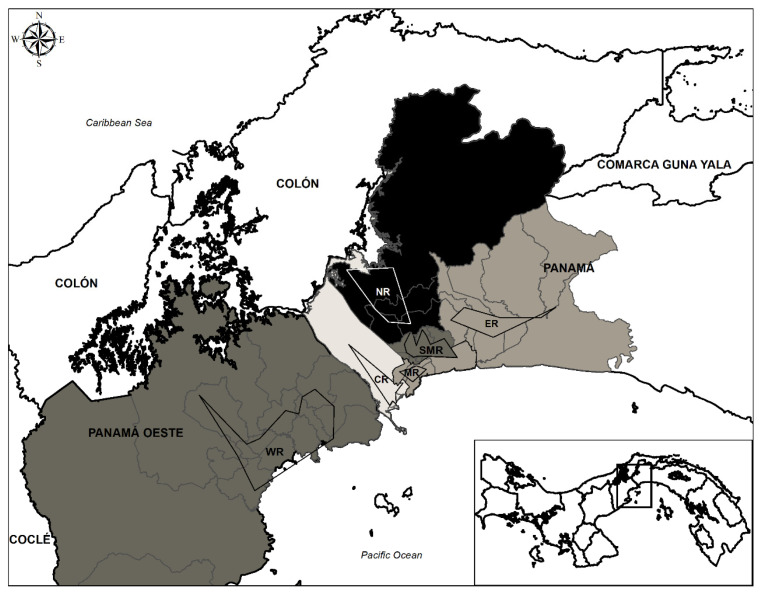
Map of Panama and West Panama. The areas of sampled regions are delimited using polygons. Sampled regions are coded as follows: NR = North region, WR = West region, SMR = San Miguelito region, ER = East region, MR = Metro region and CR = Central region. Regions are colored in a gray scale according to their *T. gondii* infection percentages and mean ± standard deviation infection percentage ranges as follows: NR (27.94–31.62); WR, SMR (24.26–27.94); ER (20.58–24.26); and CR (16.89–20.57).

**Figure 2 microorganisms-08-00927-f002:**
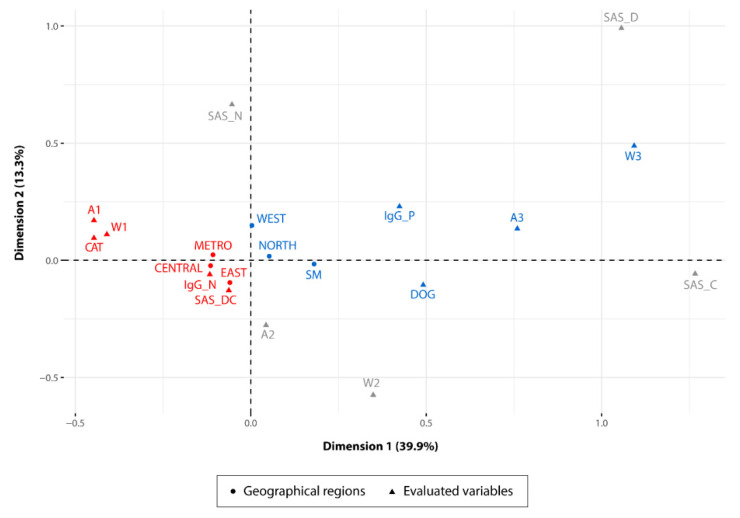
Multiple correspondence analysis (MCA). The two-dimensional diagram helps to visualize the cluster of correlated variable categories in groups. Dimensions 1 and 2 constitute the space where variable categories are expressed and explain 53.2% of variance in the dataset. Variable categories analyzed: (IgG_P) IgG positivity; (IgG_N) IgG negativity; (DOG) animal species studied: dog; (CAT) animal species studied: cat; (A1) age group 1; (A2) age group 2; (A3) age group 3; (W1) weight group 1; (W2) weight group 2; (W3) weight group 3; (SAS_N) presence of other stray animal species: none; (SAS_D) presence of other stray animal species: dog; (SAS_C) presence of other stray animal species: cat; (SAS_DC) presence of other stray animal species: dog and cat; and regions sampled, (METRO) metropolitan region, (CENTRAL) central region, (SM) San Miguelito district, (WEST) west region, (NORTH) north region, (EAST) east region. Clusters of variables associated with IgG positivity or negativity are colored in blue and red respectively. Age and weight groups are coded as follows: age group 1 = 0–5 months, age group 2 = 6–12 months, age group 3 = 13 months or older, weight group 1 = 0–5 pounds, weight group 2 = 6–12 pounds, weight group 3 = 13 pounds or more.

**Table 1 microorganisms-08-00927-t001:** Frequency of *Toxoplasma gondii* infection in stray dogs and cats of Panama City and West Panama ^1,2^.

Regions	Communities	Nº. of Stray Dogs and Cats Tested	Nº. (%) and (95% CI) of Positive Dogs and Cats	Species Analyzed	Nº. of Species Analyzed/Dogs and Cats	Nº. (%) and (95% CI) of Dogs and Cats with					Mean ± S.E.M. of Age (Months)
						Antibodies against *T. gondii*	IgG + and IgM-	IgG+ and IgM+	IgG- and IgM+	IgM+ Total	
Central	Bella Vista, Curundú, Ancón, Calidonia, San Felipe, Santa Ana, Chorrillo.	138	24 (17.39) * (11.67–24.97)	Dogs	44	11 (25.00) (13.70–40.65)	11 (25.00) (13.70–40.65)	0 (0) (0–10)	0 (0) (0–10)	0 (0) (0–10)	28.75 ± 5.96
				Cats	94	13 (13.82) (7.86–22.85)	12 (12.76) (7.06–21.62)	1 (1.06) (0.05–6.62)	0 (0) (0–4.89)	1 (1.06) (0.05–6.62)	5.58 ± 0.49
Metro	San Francisco, Pueblo Nuevo, Betania, Rio Abajo, Parque Lefevre, Juan Díaz.	115	25 (21.73) (14.81–30.60)	Dogs	40	10 (25.00) (13.24–41.52)	10 (25.00) (13.24–41.52)	0 (0) (0–10.91)	0 (0) (0–10.91)	0 (0) (0–10.91)	15.92 ± 3.51
				Cats	75	15 (20.00) (11.98–31.15)	13 (17.33) (9.90–28.18)	2 (2.66) (0.46–10.17)	0 (0) (0–6.07)	2 (2.66) (0.46–10.17)	7.16 ± 0.92
East	Las Mañanitas, Pacora, Tocumen, 24 de Diciembre, Pedregal, San Martin.	90	21 (23.33) (15.33–33.65)	Dogs	36	6 (16.67) (6.96–33.47)	6 (16.67) (6.96–33.47)	0 (0) (0–12.01)	0 (0) (0–12.01)	0 (0) (0–12.01)	13.89 ± 3.21
				Cats	54	15 (27.77) (16.86–41.86)	6 (11.11) (4.60–23.31)	4 (7.40) (2.40–18.74)	5 (9.25) (3.46-21.06)	9 (16.66) (8.36–29.79)	8.55 ± 1.38
San Miguelito	Mateo Iturralde, José Domingo Espinar, Victoriano Lorenzo, Amelia Denis de Icaza, Arnulfo Arias, Belisario Porras, Belisario Frias, Omar Torrijos, Rufina Alfaro.	155	41 (26.45) (19.85–34.24)	Dogs	95	25 (26.31) (18.05–36.52)	23 (24.21) (16.26–34.28)	2 (2.10) (0.36–8.12)	0 (0) (0–4.84)	2 (2.10) * (0.36–8.12)	14.07 ± 1.18
				Cats	60	16 (26.66) (16.45–39.89)	5 (8.33) (3.11–19.12)	2 (3.33) (0.58–12.55)	9 (15) (7.50–27.08)	11 (18.33) * (9.93–30.85)	7.57 ± 1.06
West	Barrio Balboa, Barrio Colón, Guadalupe, Playa Leona, Arraiján, Burunga, Juan Demóstenes Arosemena, Veracruz, Vista Alegre, Campana, Herrera, Nuevo Chorrillo.	75	20 (26.66) (17.42–38.34)	Dogs	46	14 (30.43) (18.20–45.92)	12 (26.08) (14.75–41.41)	2 (4.34) (0.76–16.04)	0 (0) (0–9.60)	2 (4.34) (0.76–16.04)	14.35 ± 3.67
				Cats	29	6 (20.68) (8.70–40.26)	6 (20.68) (8.70–40.26)	0 (0) (0–14.56)	0 (0) (0–14.56)	0 (0) (0–14.56)	7.96 ± 1.88
North	Alcalde Díaz, Chilibre, Las Cumbres, Ernesto Córdoba Campos, Caimitillo.	97	28 (28.86) * (20.34-39.09)	Dogs	58	16 (27.58) (17.05–41.11)	11 (18.96) (10.28–31.81)	5 (8.62) (3.22–19.72)	0 (0) (0–7.74)	5 (8.62) (3.22–19.72)	10.43 ± 1.81
				Cats	39	12 (30.76) (17.55–47.73)	8 (20.51) (9.87–36.94)	4 (10.25) (3.34–25.16)	0 (0) (0–11.17)	4 (10.25) (3.34–25.16)	9.33 ± 1.65
Total		670	159 (23.73) (20.59–27.17)	Dogs	319	82 (25.70) (21.07-30.93)	73 (22.88) (18.47–27.97)	9 (2.82) (1.38–5.47)	0 (0) (0–1.48)	9 (2.82) * (1.38–5.47)	15.69 ± 1.26
				Cats	351	77 (21.93) (17.79–26.71)	50 (14.24) (10.85–18.44)	13 (3.70) (2.07–6.41)	14 (3.98) (2.28–6.76)	27 (7.69) * (5.22–11.12)	7.33 ± 0.44

^1^ Values marked with asterisks (*) indicate statistically significant differences between the sampled regions. ^2^ Nº indicates the number of samples analyzed.

**Table 2 microorganisms-08-00927-t002:** Frequency of *T. gondii* in feral cats associated with co-infection with immunosuppressive diseases ^1^.

Regions	Communities		Nº. (%) and (95% CI) of Positive Cats with						
		Nº. of Cats Analyzed	Antibodies against *T. gondii*	Immunosuppressive Diseases	FIV	FeLV	FIV/FeLV	Immunosuppressive Diseases and *T.gondii*	Mean ± S.E.M. of Age (Months)
Central	Bella Vista, Curundú, Ancón, Calidonia, San Felipe, Santa Ana, Chorrillo.	94	13 (13.82) (7.86–22.85)	12 (12.76) (7.06–21.62)	9 (9.57) (4.74–17.85)	1 (1.06) (0.05–6.62)	2 (2.12) (0.37–8.21)	7 (7.44) (3.30–15.24)	5.58 ± 0.49
Metro	San Francisco, Pueblo Nuevo, Bethania, Rio Abajo, Parque Lefevre, Juan Díaz.	75	15 (20.00) (11.98–31.15)	13 (17.33) (9.90–28.18)	12 (16.00) (8.89–26.67)	0 (0) (0–6.07)	1 (1.33) (0.06–8.21)	7 (9.33) (4.15–18.85)	7.16 ± 0.92
East	Las Mañanitas, Pacora, Tocumen, 24 de Diciembre, Pedregal, San Martin.	54	15 (27.77) (16.86–41.86)	7 (12.96) (5.80–25.52)	6 (11.11) (4.60–23.31)	1 (1.85) (0.09–11.18)	0 (0) (0–8.27)	1 (1.85) (0.09–11.18)	8.55 ± 1.38
San Miguelito	Mateo Iturralde, José Domingo Espinar, Victoriano Lorenzo, Amelia Denis de Icaza, Arnulfo Arias, Belisario Porras, Belisario Frias, Omar Torrijos, Rufina Alfaro	60	16 (26.66) (16.45–39.89)	4 (6.66) (2.16–17.00)	3 (5.00) (1.30–14.82)	1 (1.66) (0.08–10.14)	0 (0) (0–7.50)	2 (3.33) (0.58–12.55)	7.57 ± 1.06
West	Barrio Balboa, Barrio Colón, Guadalupe, Playa Leona, Arraiján, Burunga, Juan Demóstenes Arosemena, Veracruz, Vista Alegre, Campana, Herrera, Nuevo Chorrillo.	29	6 (20.68) (8.70–40.26)	3 (10.34) (2.71–28.50)	2 (6.89) (1.20–24.21)	0 (0) (0–14.56)	0 (0) (0–14.56)	1 (3.44) (0.18–19.63)	7.96 ± 1.88
North	Alcalde Díaz, Chilibre, Las Cumbres, Ernesto Córdoba Campos, Caimitillo.	39	12 (30.76) (17.55–47.73)	1 (2.56) (0.13–15.08)	2 (5.12) (0.89–18.63)	0 (0) (0–11.17)	0 (0) (0–11.17)	0 (0) (0–11.17)	9.33 ± 1.65
Total		351	77 (21.93) (17.79-26.71)	40 (11.39) (8.36–15.30)	34 (9.68) (6.89–13.39)	3 (0.85) (0.20–2.69)	3 (0.85) (0.20–2.69)	18 (5.12) * (3.16–8.13)	7.33 ± 0.44

^1^ Values marked with asterisks (*) indicate statistically significant differences between the sampled regions. Nº indicates the number of samples analyze.

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
