# Peer review of "Frequency of Toxoplasma gondii and Risk Factors Associated with the Infection in Stray Dogs and Cats of Panama"

_microorganisms, 2020, doi:10.3390/microorganisms8060927_

Round 1
Reviewer 1 Report
I think that the manuscript needs to be improved and maybe a short communication would be more appropriate. I also suggest to send it to a more regional journal because it is well known that stray cats and dogs might represents a risk for T. gondii transmission and diffusion of the parasite in the environment, in this way you can mark the need for a national plan in reducing the number of stray cats and dogs in the country. I would like also to suggest a future genotyping of the parasite in the country, no recent data are available in literature.
Author Response
Dear Reviewer,
I agree that the manuscript needed to be improved and we worked to make all the changes suggested in the recommendations. We believe that after making these corrections, the manuscript has reached the academic level required for it to be published in this prestigious journal.
Although there is considerable research worldwide on this topic, in the Latin American region there are few studies focused on populations of stray dogs and cats. Most of the research has been done on pets and only in some countries in this region. Therefore, the data obtained in this study not only provides important information at the regional level, but also contributes with relevant information that can be compared with data from other countries.
On the other hand, in this study important epidemiological data were obtained. The dynamics of the infection between the studied regions was also determined through in-depth statistical analyzes. In addition, specific variables of each region possibly associated with the transmission of the parasite were observed. We also carry out analyzes with other infections (feline immunodeficiency virus and feline leukemia virus) to assess the level of association that these diseases can have with T. gondii infection. Therefore, we think it is a fairly comprehensive study that may be relevant from a public health point of view.
As you well pointed out, this type of study has the potential to highlight the need to implement a national plan with public policies that promote controlled sterilization that will help to reduce the population of stray animals. Implementing educational programs on responsible pet ownership could also help to prevent the abandonment of these animals and the spread of zoonotic diseases.
Regarding the analysis of genotypes, we are trying to obtain funds to study the genetic variability of T. gondiiboth in domestic animals and in wild animals. In Panama, no data has yet been reported on the genotypes that may be circulating in urban, rural and jungle regions. Therefore, it is extremely important that these studies are carried out and can be compared with the genotypes currently reported by other countries. The publication of the current manuscript might increase the chances of getting funded by the National Secretary of Science, Technology and Innovation.
Thank you very much for your suggestions
Reviewer 2 Report
The manuscript “Frequency of Toxoplasma gondii and risk factors associated with the infection in stray dogs and cats of Panama” is an interesting report talking about the dynamics of Toxoplasma gondii infection in stray dogs and cats from two major regions of Panama in six different sub-populations. I believe that studies with wild/feral animals are important since there is a lack of studies of this type to understand the disease dynamic as a whole in some countries.
Here are some comments:
- Abstract: I would change “Therefore, the objective of this study was to evaluate the frequency of T. gondii in stray dogs and cats in Panama City and West Panama and determine risk factors associated ….” for “Therefore, the objective of this study was to evaluate the frequency of T. gondii in stray dogs and cats in six different regions from Panama and determine risk factors associated...”
- Line 167 – Reference protocol methodological
- Suggestion: 2.4 and 2.5 could be merged into one section
- Add the exact number of samples collected in the methods section (319 dogs and 351 cats)
- Line 218 – remove “.” after chance
- Line 224 – “The average age estimated for ...”
- Figure 1 – work on colors, the light blue from ER and SMR are almost identical to the light blue from the ocean. Maybe maintaining the ocean as white should fix the problem.
- For each region studied, besides having a considerable close average number of samples, we can observe that the number of communities studied varies, should this affect the diversity analyzed of each one of the six sub-regions? How these samples are distributed among these communities? The group shows in table 1 and two the distribution on each region studied, but how they are distributed among dogs and cats separately on each community studied?
- Figure 2 should have more graphical information (color legend, demarcate clusters in MCA) It has a lot of information in the figure description but is hard to follow in the figure.
- What happens when comparing these results to domestic animals? Is there any correlation visible?
Author Response
Dear Reviewer, the following text contains the answers to your questions.
Abstract:
- On line 32 of the document in word format, the suggested changes were made.
Methodology
- In line 183 of the document in word format, the required references were added. This line corresponds to line 167 in pdf format.
- I understand your suggestion regarding the collection of data and information, however, I think it is better to leave both methodologies separate for a better understanding of the results.
- In line 188 of the document in word format, the total samples collected for the population of dogs (319) and cats (351) were added.
- Line 235 that corresponds to line 224 of the document in word format the "." was removed after the word chance.
Results
- Line 243 that corresponds to line 231 of the document in word format, the word "estimated" was added.
- Figure 1: The colors of the oceans were changed to white and were delimited by lines and labels. The colors of the regions were changed to black and white colors by recommendation of the reviewer 3. A degradation of the black color was used, showing the areas of higher prevalence in darker colors than those areas of lower prevalence in lighter colors. It was also possible to show the regions with similar prevalence as East and Metro and as the San Miguelito and West regions. On the other hand, the previous figure had an error for the SMR and MR regions that did not agree with the data in Table 1. Thanks to their recommendations, we were able to notice the error and correct it in the new figure. I apologize for the previous errors in figure 1.
- The communities shown in Tables 1 and 2 represent the provenance of the samples within the analyzed regions. The East, Central, Metro and North regions have a homogeneous number of samples per community and the distribution of the samples of dogs and cats remain fairly homogeneous within each of the communities sampled. However, for the San Miguelito and West Regions where the number of communities is large, we have communities that do not meet the required sample size. For the San Miguelito region we have 3 communities that are below the minimum size, this fact was also observed in the dog and cat populations. The same was observed for 4 regions of the West Region. It is important to mention that each one of the studied regions varies in surface (Km2), population density and number of communities.
For this reason, we tried to obtain samples representing the majority of the communities, mainly for the San Miguelito and West Regions, which had the highest number of communities per Region. Although we have these discrepancies in terms of the number of samples per community, the spatial distribution of them remains fairly homogeneous within each of the sampled regions. In addition, the sample size obtained for the 6 regions studied is above the minimum number of samples required, therefore, the diversity analyzed in these regions must be representative of the total population of animals.
- Figure 2: The clusters that formed in the MCA were defined by the red and blue colors. The red color represents the variables most associated with IgG negativity. The blue color represents the variables most associated with IgG positivity. The geographic regions were defined by points and the variables by triangles.
- A Chi square test was performed (with a significance level of 0.05) to compare the results of both studies. This test was only significant when the percentages of the East regions were compared (23.33% for domestic animals and 39.56 for stray animals). The Chi square value was 14.02, 4 degrees of freedom and a P value of 0.0072. The San Miguelito and West regions showed similar percentages for both studies. The Central region showed some discrepancies in the communities sampled between both studies, therefore, it was not considered to carry out the comparative analyzes.
Reviewer 3 Report
Work presented by Fabrega and collages describes Frequency of T. gondii and risk factors associated with the infection in stray dogs and cats of Panama. The manuscript touches a significant problem that is a role of homeless animals. This study represents a significant contribution to the field of veterinary parasitology and public health.
However, I have some minor reservations as follows.
Introduction:
The introduction is well written however, the paragraph from line 85 to 125 is a little bit too long. I would consider shortening this section.
Results:
The statistical approach needs to be clarified:
*Prevalence values should be reported with 95% Confidence intervals - this can be easily calculated using SPPS software. See Zaloumis et al. (2015): Presenting parasitological data: the good, the bad and the error bar. Parasitology http://dx.doi.org/10.1017/S0031182015000748
*Please provide statistical values for given results - X2, degrees of freedom, p values - it is really important in analysing obtained results. Please see Morrison D.A.(2002): How to improve statistical analysis in parasitology research publications. International Journal for Parasitology 32, 1065-1070.
*Line 222 The percentage of positivity for T. gondii -> please change into Seroprevalence or Prevalence of antibodies against T. gondi.
* Line 225 – please change average into mean. If you report a mean please add S.E.M
* Line 225-228 and Figure 1. Please rewrite this section. It is not clear which region had higher seroprevalence. Figure 1 is hard to read. I would change it for a clear graph rather map with colours. Moreover, having a colour Map showing results may be a problem for readers who print the article in black and white.
*Line 224-224 please provide statistical values.
Since English is not my first language, I usually refrain from advising text editing. However, I find this manuscript to be clear and well written. I have found only a few typos that might be corrected during the editorial process.
Overall, I recommend the article for publication after amending it according to above comments.
Author Response
Dear Reviewer, the following text contains the answers to your questions.
Introduction:
1) Line 85 to 125: I agree that this part is a bit long, however, I consider it important to highlight the few data that exist on the subject in Latin America. In addition to the data from these works presented in the introduction, they complement the results obtained in this research and call attention to further research that helps to understand the true role of these animals in the transmission of the parasite.
Results
1) Confidence intervals were calculated for all percentages of positivity and placed in Tables 1 and 2.
2) Statistical values of X2, degrees of freedom and the value of P were placed in the text.
3) Line 222: the percentage of positivity for T. gondii was changed to the word Seroprevalence. The line number changed to 239 in the word format document.
4) Line 225: the average word was changed to the mean word. The line number changed to 243 in the word format document. The means corresponding to the age of the animals were placed in the text and in Table 1 together with their respective standard error of the mean (SEM). Thanks to his recommendations for calculating the standard error of the mean, we could see that the means of the ages of the animals placed in tables 1 and 2 of the manuscript were wrong. Those values were calculated at the beginning of the sampling and were not updated. All means of the ages of the animals were updated in both tables.
5) Lines 225 and 228 were rewritten and it was clarified which is the region with the highest seroprevalence. These lines in the word format document correspond to lines 239 to 249. I apologize because the map had an error in the figure and in the legend. Thanks to your reviews we were able to notice this error. The map colors were changed to black and white and a gray gradient. With the corrected map, I think that it is possible to visualize more clearly the regions with the highest prevalence than those regions with the lowest prevalence. I think that it would no longer be necessary to place a graph also because the seroprevalence data is placed in Table 1. The map was placed with the aim of showing the location of the studied regions, the sampling area across the polygons and how the prevalence is distributed in these regions.
6) For line 224 the statistical values such as the P value and the degrees of freedom were added to the text. This line in the document in word format corresponds to line 241.

Reviewer 4 Report
Dear authors,
Thank you for submitting the manuscript.
After careful reading, the overall manuscript is interesting which highlights the seroprevalence of T. gondii in fiction in stray dogs and cats that may post a higher chance of zoonotic transmission to humans.
I would suggest the following to improve the current version of the manuscript.
- All highlighted of infection, please revise either the authors want to add as T. gondii or Toxoplasma (italic) as appropriate.
- In full references, please italic all scientific name(s) as highlighted in yellow colors.
- I think the overall results are about T. gondii infection NOT toxoplasmosis (disease/case etc) so please revise as appropriate to avoid confusion. All are in yellow highlighted colors.
- Few more comments as appeared in all yellow highlighted colors.

Author Response
1. All highlighted of infection, please review either the authors want to add as T. gondii or Toxoplasma (italic) as appropriate.
Answer: The highlights were placed as T. gondii infection.
2. In full references, please italic all scientific name (s) as highlighted in yellow colors.
Answer: In the references section, all scientific names were written in italics. The references were corrected from the EndNote format, therefore the tracking changes are not reflected.
3. I think the overall results are about T. gondii infection NOT toxoplasmosis (disease / case etc) so please check as appropriate to avoid confusion. All are in yellow highlighted colors.
Answer: The indicated highlights were corrected by T. gondii infection.
4. Few more comments as appeared in all yellow highlighted colors. Answer: All comments were corrected. However, the indication made in the keywords is not clear. Please clarify what the correction is at this point.
The document was corrected in word format with all the indications of the reviewer, please review the annex.
Round 2
Reviewer 1 Report
I suggest to reduce the paraghaph from line 88 to 133, it is too long and I think that the number of examples reported could be reduced or in any case resumed in few sentences.
Author Response
Dear reviewer, we accept your suggestions and the text in the introduction section from line 88 to line 133, was reduced into shorter sentences however we try to keep the essence of the initial content. In annex are the modified paragraphs.
Epidemiological studies in Latin America Countries have shown a high frequency of T. gondii infection in stray dogs and cats strongly associated with the region, the animal species and the type of diagnostic test used. Studies conducted in Brazil displayed variations in urban regions of the cities of Sao Paulo (dogs-50.5%, cats-40.0%) Rio de Janeiro (cats-18%) and Curitiba (dogs-48%), where a high percentage of positivity was observed in both dogs and cats [16] [17] [18].
In Argentina and Chile, the majority of the reports are studies carried out on pets. Different ranges of seroprevalences have been found in regions such as the Argentine Chaco (dogs-55.3%), Buenos Aires (cats- 19.5% to 22.6%) and Northeast Argentina (dogs-13.1% to 23.0%) [19] [20] [21] [22]. In Chile, the Valdivia and San Carlos regions yielded seroprevalence data of T. gondiiinfection in cats of 33.0% and 48.3%, respectively [23] [24]. Another study in southern Chile showed a high prevalence for T. gondii infection in a population of domestic and feral cats (68%) [25]. Unfortunately, the data was not stratified, therefore no inferences can be made about risk factors among these populations.
Furthermore, in the cities of Bogota and Lima the frequencies of T. gondii infection in abandoned dogs and domestic cats was 16.8% and 11.0% to 17.9% respectively [26] [27] [28].
In countries of the Caribbean, Central America and Mexico, few studies have been carried out on stray dogs and cats. However, high frequencies of infection in feral cats from an island of Puerto Rico (84.2%) have been reported [29]. In different regions of Mexico and Panama, seroprevalence data were only found in domestic animals, with ranges between 70.8% to 91.8% and 9.2% to 42.0% respectively [30] [31] [32] [33]. Furthermore, a recent study carried out in regions of Panama City and Panama West reported high seroprevalences in domestic dogs and cats (30.73% of overall prevalence) without significant differences between the populations of both species [34].
Reviewer 2 Report
The group did a great job in the reviewed manuscript. I would suggest adding some of the missing information provided by review section 8 in the review (attached below) showing some of these observations (some of them are already present, but things like each community distribution and explanation should be nice to have added). The only thing that I need from you is to add some explanation justifying the reason used for showing the sampling in that way and what impact (or not) this could cause to the analysis.
"The communities shown in Tables 1 and 2 represent the provenance of the samples within the analyzed regions. The East, Central, Metro and North regions have a homogeneous number of samples per community and the distribution of the samples of dogs and cats remains fairly homogeneous within each of the communities sampled. However, for the San Miguelito and West Regions where the number of communities is large, we have communities that do not meet the required sample size. For the San Miguelito region, we have 3 communities that are below the minimum size, this fact was also observed in the dog and cat populations. The same was observed for 4 regions of the West Region. It is important to mention that each one of the studied regions varies in surface (Km2), population density, and the number of communities.
For this reason, we tried to obtain samples representing the majority of the communities, mainly for the San Miguelito and West Regions, which had the highest number of communities per Region. Although we have these discrepancies in terms of the number of samples per community, the spatial distribution of them remains fairly homogeneous within each of the sampled regions. In addition, the sample size obtained for the 6 regions studied is above the minimum number of samples required, therefore, the diversity analyzed in these regions must be representative of the total population of animals."
Other than that they covered everything that I could see from my comments and the other reviewers.
Author Response
Dear reviewer, we accept your suggestions and a paragraph was added in the methodology section (lines 319 to 328), which explains the distribution of the communities and the criteria used to carry out the sampling in these communities.
The samples collected in this study come from 46 communities distributed as follows: 7 communities in the Central Region, 6 in the Metro Region, 6 in the East Region, 9 in the San Miguelito Region, 12 in the West Region and 6 in the North Region. Some communities within the San Miguelito and West regions showed variations in their sample size. However, this do not affect inter-regional analysis given that the spatial distribution of the samples collected in each of the six analyzed regions remained homogeneous, and that the sampling was based on the areas with the highest human population density where feral dogs and cats have the highest number of interactions. The communities described in Tables 1 and 2 only reflect the provenance of the samples and it is through the confidence intervals shown in the tables it was possible to observe the range of variation of the percentages of positivity found in each of the studied regions.